# Fabry Disease: Switch from Enzyme Replacement Therapy to Oral Chaperone Migalastat: What Do We Know Today?

**DOI:** 10.3390/healthcare11040449

**Published:** 2023-02-04

**Authors:** Fernando Perretta, Sebastián Jaurretche

**Affiliations:** 1Unidad de Terapia Intensiva, Hospital Dr. E. F. Erill de Escobar, Escobar B1625, Provincia de Buenos Aires, Argentina; 2Fresenius Medical Care Escobar, Escobar B1625, Provincia de Buenos Aires, Argentina; 3Trasplante Renal-Pancreático, Sanatorio Parque de Rosario, Rosario S2000, Provincia de Santa Fe, Argentina; 4Biofísica y Fisiología Humana, Escuela de Medicina, Instituto Universitario Italiano de Rosario, Rosario S2000, Provincia de Santa Fe, Argentina

**Keywords:** Fabry disease, globotriaosylceramide, α-galactosidase-A, enzyme replacement therapy, chaperone therapy, migalastat

## Abstract

Fabry disease is a lysosomal storage disorder caused by the deficiency of the α-galactosidase-A enzyme. The result is the progressive accumulation of complex glycosphingolipids and cellular dysfunction. Cardiac, renal, and neurological involvement significantly reduces life expectancy. Currently, there is increasing evidence that clinical response to treatment improves with early and timely initiation. Until a few years ago, treatment options for Fabry disease were limited to enzyme replacement therapy with agalsidase alfa or beta administered by intravenous infusion every 2 weeks. Migalastat (Galafold^®^) is an oral pharmacological chaperone that increases the enzyme activity of “amenable” mutations. The safety and efficacy of migalastat were supported in the phase III FACETS and ATTRACT studies, compared to available enzyme replacement therapies, showing a reduction in left ventricular mass, and stabilization of kidney function and plasma Lyso-Gb3. Similar results were confirmed in subsequent extension publications, both in patients who started migalastat as their first treatment and in patients who were previously on enzyme replacement therapy and switched to migalastat. In this review we describe the safety and efficacy of switching from enzyme replacement therapy to migalastat in patients with Fabry disease and “amenable” mutations, referring to publications available to date.

## 1. Introduction

Fabry disease (FD, OMIM 301500) is a rare, X-linked, lysosomal storage disorder caused by the absence or deficiency of the α-galactosidase-A enzyme (α-gal-A, EC 3.2.1.22) [1]. The result is complex glycolipid accumulation in body fluids and different tissues, mainly globotriaosylceramide (GL-3 or Gb3) and globotriaosylsphingosine (Lyso-Gb3) [1,2]. FD can be classified into a classic and a non-classic or later-onset phenotype also called an adult variant. Patients with the classic disease have undetectable or very low enzyme activity (<3% of normal value) and develop organic complications early in life, including stroke, hypertrophic cardiomyopathy, and renal failure [2,3]. The disease in women may be more variable, from practically asymptomatic to severely affected as in men, depending in part on the mutation type, the residual α-gal-A enzyme activity level, and the pattern of X chromosome inactivation (Lyon hypothesis). Epigenetic factors may also contribute to the clinical variability in female patients [4,5,6]. The later-onset phenotypes are associated with higher residual enzyme activity, meaning these patients are generally less affected, with manifestations limited to a single organ, such as the cardiac or renal variant [2,3]. For male patients, α-gal-A activity testing is diagnostic, and the confirmation of the mutation is important to determine the disease phenotype. In female patients, the presence of a GLA gene mutation is required as the plasma enzyme activity is often within the normal range [2]. Recent analysis of international newborn screening evidenced frequencies of 1:22,570 males for the classic phenotype and 1:1390 males for the non-classic or later-onset phenotype, which would position this pathology as the most frequent lysosomal storage disease [7,8].

Current therapeutic options include enzyme replacement therapy (ERT), agalsidase alfa (Replagal^®^, Takeda-Shire Human Genetic Therapies, Lexington, MA, USA), and agalsidase beta (Fabrazyme^®^, Sanofi-Genzyme, Cambridge, MA, USA), administered by intravenous infusion. Agalsidase alfa is used at a licensed dose of 0.2 mg/kg body weight every other week (EOW) and agalsidase beta at 1.0 mg/kg body weight EOW [9,10]. During the last 20 years, different studies have demonstrated the efficacy of ERT in different organ systems affected by FD. A novel alternative that overcomes some limitations of ERT, the pharmacological chaperone migalastat (Galafold^®^, Amicus Therapeutics, Cranbury, NJ, USA) was approved in the European Union in 2016 and the United States in 2018 for patients with FD and “amenable” mutations [11]. Migalastat is an imino-sugar that selectively and reversibly binds to “amenable” mutant forms of the enzyme α-gal-A, facilitating trafficking to lysosomes to metabolize the accumulation of Gb3 [11,12]. It is estimated that between 35 and 50% of FD mutations are “amenable” to migalastat [13]. The efficacy of the pharmacological chaperone migalastat was evaluated in two pivotal multicenter phase III studies, FACETS [14] and ATTRACT [13], and in subsequent open-label extension studies. This review aims to describe the safety and efficacy of switching from enzyme replacement therapy to migalastat in FD patients and “amenable” mutations, referring to publications available to date.

### Chaperone Therapy in Fabry Disease

Migalastat is a low-molecular-weight imino-sugar that binds selectively and reversibly to the active sites of “amenable” mutant forms of α-gal-A enzyme [15]. This binding stabilizes it and prevents its degradation in the endoplasmic reticulum and facilitates trafficking to lysosomes, where migalastat dissociates from α-gal-A, permitting the enzyme to degrade GL-3 [16]. The absolute bioavailability of migalastat after a single dose is ≈75%, and is extensively distributed with an apparent volume of distribution of ≈89 L (range 77–133 L). Migalastat is rapidly removed from the plasma compartment (mean elimination half-life of ≈4 h). In healthy volunteers, 77 and 20% of the total migalastat radiolabeled dose was recovered in urine and faeces [11]. The amenability to migalastat is determined by a validated good laboratory practice (GLP) in vitro assay with transfected human embryonic kidney (HEK) cells. Migalastat “amenable” mutations are defined by α-gal-A activity that is ≥1.20-fold over the baseline with an absolute increase of ≥3.0% wild-type α-Gal-A activity in the presence of 10 μmol/L migalastat. The amenability is applicable in male and female patients and does not require patient samples [17]. More than 1000 FD mutations have been described, with about 30–50% considered “amenable” to migalastat therapy (mainly missense mutations) [13].

The suggested dosage of migalastat is 123 mg once every other day at the same time each day on an empty stomach (no food should be eaten for at least 2 h before and 2 h after migalastat administration). In patients with severe renal failure (estimated glomerular filtration rate (eGFR) < 30 mL/min/1.73 m^2^), migalastat is not recommended, including patients on dialysis. Furthermore, migalastat should not be administered concomitantly with ERT [11,12]. Local prescribing information should be consulted for more details, precautions, contraindications, and special populations.

The benefits of migalastat treatment in FD patients with “amenable” mutations were demonstrated in two pivotal, randomized, multicenter, placebo-controlled (FACETS [14]) or active-comparator-controlled (ATTRAC T [13]) phase III trials, and in subsequent open-label extension studies. In summary, these trials evidenced a significant reduction in left ventricular mass index (LVMi) and stabilization of renal function, and diarrhea as an FD symptom was improved with migalastat treatment. In addition, the biomarker plasma Lyso-Gb3 decreased in previously untreated patients and remained stable in ERT-pretreated patients. Migalastat was generally well-tolerated in patients with Fabry disease and “amenable” mutations. The most common adverse effects reported were headaches and nasopharyngitis [13,14]. In recent years, the indication for migalastat as an oral monotherapy for FD has been increasing worldwide, not only in treatment-naive patients but also in patients who switch from ERT to migalastat.

## 2. Switching from Enzyme Replacement Therapy to Migalastat

After a bibliographic search in different electronic databases (MEDLINE, EMBASE, SCOPUS, Cochrane, Latindex, and Google Academic) regarding the switch from ERT to the pharmacological chaperone migalastat, five publications were found. The terms used for searching were “Fabry disease”, “enzyme replacement therapy”, “migalastat”, and “switch”. The characteristics of these publications are shown in Table 1.

In the phase III ATTRACT study [13], the principal objective was to evaluate migalastat effects on kidney function among FD “amenable” patients previously treated with ERT (agalsidase alfa 0.2 mg/kg or agalsidase beta 1 mg/kg), along with cardiac effects, disease substrate, patient-reported outcomes, and safety. A total of 60 patients aged 18–72 years (56% female) were randomized; for different reasons, only 52 patients (34 in the group that switched from ERT to migalastat and 18 in the group that remained on ERT) finished the 18-month randomization phase. Most of the patients included in this study had multiple-organ involvement through FD, considering their baseline characteristics and medical reports. The kidney results demonstrated that migalastat has a similar effect to ERT, stabilizing renal function in terms of glomerular filtration rate (GFR) loss and proteinuria. The mean change from the baseline in 24 h urine protein was lower for the migalastat group than the ERT group after 18 months (49.2 vs. 194.5 mg). Regarding cardiac outcomes, the results showed a significant decrease in LVMi from the baseline to month 18 in FD patients treated with migalastat (−6.6 g/m^2^; 95% CI −11.0 to −2.2). In the analysis of events, the frequency in the migalastat group was 29% versus 44% in the ERT group. Patients experienced no changes in their quality of life after the switch from ERT to migalastat, and the SF-36 v2 and Brief Pain Inventory scores remained stable during the study for both treatment groups. The results also showed stabilization of plasma Lyso-Gb3 among migalastat group patients. Finally, migalastat was generally found to be safe and well-tolerated throughout the ATTRACT study.

In 2019, Müntze et al. published the first real-world data on using chaperone therapy with migalastat to treat FD patients [18]. In this prospective single-center study, migalastat’s efficacy and biomarkers changes were evaluated after 1 year of treatment for 14 FD patients (mean age of 55 ± 14 years); six of these patients were from the group that switched from ERT (agalsidase alfa 0.2 mg/kg or agalsidase beta 1 mg/kg) to migalastat. Regarding Fabry-specific biomarkers, both female and male patients evidenced a significant α-gal-A activity increase (0.06–0.2 nmol/minute/mg protein; *p* = 0.001), and plasma Lyso-Gb3 was decreased in treatment-naive patients (10.9–6.0 ng/mL; *p* = 0.021) and stable in patients who switched from ERT to the pharmacological chaperone migalastat (9.6–12.1 ng/mL; *p* = 0.607). After 1 year of migalastat treatment follow-up, a significant reduction in LVMi (137–130 g/m^2^; *p* = 0.037) was observed, and biomarkers hs-troponin T and NT-ProBNP remained stable. This cardiac hypertrophy reduction was associated with higher α-gal-A activity (r = −0.546; *p* = 0.044). In this study, kidney function decreased after 1 year of migalastat, and creatinine increased from 0.94 to 1.0 mg/dL (*p* = 0.021) in the total group, but part of the cohort started migalastat and angiotensin-converting enzyme inhibition simultaneously. Moreover, it is known that renal stabilization requires more than 1 year, and renal function can reduce in a highly heterogeneous manner [22]. Some limitations of the study were as follows: it was carried out at a single center, with only 14 FD patients, and only two follow-up analyses were performed after the initiation of migalastat treatment.

Long-term efficacy and safety of the pharmacological chaperone migalastat were reported in the open-label extension to 30 months of the phase III ATTRACT study by Feldt-Rasmussen et al. [19]. In this extension period, patients who received the pharmacological chaperone migalastat continued receiving migalastat (group 1 or MM), and patients who received ERT were switched to start migalastat treatment (group 2 or EM). A total of 46 patients who completed the ATTRACT study continued into the 12-month extension period; 31 patients in group 1 (MM) and 15 in group 2 (EM). Renal results over 30 months of treatment evidenced that eGFR remained stable with migalastat in both groups, with mean annualized rates of change of −1.7 in group 1 (MM) and −2.1 mL/min/1.73 m^2^ in group 2 (EM). In terms of the 24 h urine protein, no significant change from the baseline was observed in group 1 (MM) from 0 to 30 months, or in group 2 (EM) during either the initial ERT period or the extension migalastat period. A significant reduction in cardiac mass was observed with migalastat treatment in group 1 (MM) in patients with left ventricular hypertrophy at the baseline (*n* = 10; mean change: −10.0 g/m^2^ [median: −11.3; 95% CI: −16.6, −3.3]), but cardiac mass remained stable in group 2 (EM) during ERT treatment in this subgroup of patients. During the open-label extension period, few patients experienced new Fabry-associated clinical events and no new safety concerns were raised. To summarize, in patients with FD and “amenable” mutations, migalastat at 123 mg once every other day was well-tolerated and evidenced durable, long-term stability of kidney function and reduction in LVMi.

In 2020, Riccio et al. reported on a single-center observational study in Italy that investigated switching from ERT (agalsidase alfa 0.2 mg/kg or agalsidase beta 1 mg/kg) to migalastat in seven adult male patients (18–66 years) with “amenable” FD mutations: five classic and two later-onset variants [20]. Neurologic, cardiac, and renal function, health status, α-gal-A activity, and Lyso-Gb3 were evaluated by comparing retrospective data at the baseline (pre-ERT) and after one year of ERT with prospective data after one year of migalastat treatment. The results showed a significant improvement in LVMi (*p* = 0.028) and proteinuria (*p* = 0.048) with the pharmacological chaperone migalastat vs. ERT, and migalastat treatment led to a decrease in plasma Lyso-Gb3 levels and an increase in α-gal-A activity. Adverse effects were similar at 28% for both treatments (migalastat and ERT). Neurologic function, pain symptoms, and health status remained unchanged in this study. However, the statistical power of this research was limited because it was implemented at a single center, with a small number of patients, and with a short follow-up period. In conclusion, the switch was safe and well-tolerated in this study.

A prospective 24-month observational multicenter study with migalastat for FD patients was published by Lenders et al. last year [21]. Fifty-four adult patients (33 previously treated with ERT) were studied after 12 and 24 months of migalastat treatment, analyzing their cardiovascular and renal function, disease severity, and changes in plasma Lyso-Gb3. FD signs and symptoms remained stable (*p* > 0.05). A reduction in LVMi was observed after 24 months of migalastat treatment (all: −7.5 ± 17.4 g/m^2^, *p* = 0.0118; females: −4.6 ± 9.1 g/m^2^, *p* = 0.0554; males: −9.9 ± 22.2 g/m^2^, *p* = 0.0699), particularly in males with cardiac hypertrophy at the baseline. The strongest effect on LVMi was demonstrated in the first 12 months. Renal results evidenced moderate yearly eGFR loss in female and male patients (−2.6 and −4.4 mL/min/1.73 m^2^ per year; *p* = 0.0317 and *p* = 0.0028, respectively), mostly in those with renin-angiotensin-aldosterone system inhibitors or aldosterone receptor blockers, which indicated high baseline involvement. Specific FD scores (Disease Severity Scoring System and Mainz Severity Score Index), α-gal-A activities, and plasma Lyso-Gb3 levels persisted and remained stable during migalastat treatment, even though some male patients showed increasing Lyso-Gb3 values over time. Finally, the authors highlighted the safety of migalastat treatment, suggesting regular monitoring of the clinical response in FD patients.

## 3. Discussion

This is the first review of the safety and efficacy of switching from ERT to the oral chaperone migalastat in patients with FD and “amenable” mutations, referring to the publications available to date. The conclusions of these reports, in favor of migalastat treatment, are shown in Table 2. A total of 95 patients who switched were included in the studies analyzed (Table 1). In 2019, Hughes et al. published a research letter describing the experience gained from the Phase III ATTRACT study of switching from ERT to migalastat [23]. Since there is no consensus on when to choose migalastat over ERT, this group suggested some criteria, which include: an age of 16 years and older (check local prescribing information), a confirmed “amenable” FD mutation, an eGFR >30 mL/min/1.73 m^2^, compliance with oral administration every other day, and no intention of pregnancy in females. The authors recommended a counseling session with FD patients. The final conclusion was that patients with FD and “amenable” mutations who have received ERT can be safely switched to migalastat treatment.

A long-term follow-up of renal function in patients treated with migalastat was published in 2021 [24]. The study included 78 patients; treatment-naive (*n* = 36 [23 females]) and ERT-experienced (*n* = 42 [24 females], with "amenable" mutations who received migalastat 123 mg every other day for ≥2 years. In this post hoc analysis, the results evidenced that patients with migalastat therapy and “amenable” mutations had a stable kidney function during the follow-up (≤8.6 years), irrespective of sex, phenotype, or treatment status. The authors highlighted the importance of early treatment to stabilize or slow the decline in renal function. A recent review concluded that to date, the main useful biomarker for Fabry nephropathy monitoring in patients receiving migalastat is eGFR using equations with plasma creatinine. The biomarkers albuminuria and proteinuria could be helpful to assess the indication for concomitant treatment or kidney biopsy in selected FD patients [25]. Fabry-associated clinical events (FACEs) cause significant morbidity and mortality. A post hoc analysis evaluated the incidence of FACEs in 97 treatment-naive and ERT-experienced adults with FD and “amenable” mutations who were treated with migalastat for up to 8.6 years in phase III clinical trials [26]. A total of 22 FACEs in 17 patients were reported (48.3 events per 1000 patient-years), with no deaths; a higher incidence was observed in males vs. females, patients aged >40 years vs. younger ones, treatment-naive vs. ERT-experienced patients and males with the classic variant vs. males and females with all other phenotypes. The incidence of FACEs remained low during long-term therapy with migalastat and a lower baseline eGFR was a predictor of FACEs.

ERT was the only therapeutic alternative available for FD patients for several years. Recently, new therapeutic approaches have been introduced including chaperone therapy, substrate reduction therapy, and gene therapy. There are different reasons why the pharmacological chaperone migalastat is an attractive option for the treatment of patients with FD and “amenable” mutations. First of all, migalastat is an oral treatment that avoids intravenous ERT infusions and consequently possible associated complications (e.g., headaches, allergic reactions, anaphylaxis, etc.) [11,27]. In addition, pharmacological chaperones are not immunogenic and so are not expected to have tolerability issues similar to those described for the different recombinant enzyme therapies [28]. Moreover, as a small molecule, it has the ability to cross the blood–brain barrier in humans, as shown in mice [29,30]. Furthermore, migalastat, being an oral therapy, allows more sustained and stable α-gal-A levels than ERT [28]. Recent studies showed that in vitro and in vivo amenability may not always match in certain mutations, with an insufficient increase of α-gal-A enzymatic activity, suggesting a regular follow-up with laboratory measurements to verify clinical response [31,32]. Regarding limitations of migalastat treatment, we can mention that the chaperone cannot be prescribed with severe renal failure (eGFR < 30 mL/min/1.73 m^2^), patient’s compliance should be monitored in order to optimize its efficacy, and clinical experience with migalastat is limited compared to ERT.

Our experience in Argentina to date is gained from follow-up of a cohort of 20 patients, both with migalastat as the initial therapy and also with the switch from ERT to migalastat. No significant pharmacological adverse effects or serious clinical events (cardiac, renal, or cerebrovascular) have been reported in our population during the initial follow-up (preliminary data, not yet published). 

## 4. Conclusions

In conclusion, based on its proven efficacy in the reduction of LVMi and stabilization of renal function and disease biomarkers, migalastat offers a safe alternative for switching from ERT or initiating therapy in patients with FD and “amenable” mutations.

## Figures and Tables

**Table 1 healthcare-11-00449-t001:** Characteristics of the publications.

Authors	Single/Multi-Center	Follow-Up	*n* Total	*n* Switch	Both Genders	Classic and Later-Onset
Hughes et al. [13]	Multicenter	18 months	52	34	Yes	Yes
Müntze et al. [18]	Single-center	12 months	14	6	Yes	Yes
Feldt-Rasmussen et al. [19]	Multicenter	12 months	46	15	Yes	Yes
Riccio et al. [20]	Single-center	12 months	7	7	No	Yes
Lenders et al. [21]	Multicenter	24 months	54	33	Yes	Yes

**Table 2 healthcare-11-00449-t002:** Conclusions of the publications.

Authors	Conclusions in Favor of Migalastat Treatment
Hughes et al. [13]	Stabilization of renal function Decrease in the left ventricular mass index Stabilization of plasma Lyso-Gb3
Müntze et al. [18]	Increase in α-gal-A activity Decrease in the left ventricular mass index Stabilization of plasma Lyso-Gb3
Feldt-Rasmussen et al. [19]	Stabilization of renal function Decrease in the left ventricular mass index
Riccio et al. [20]	Increase in α-gal-A activity Stabilization of renal function Decrease in the left ventricular mass index Stabilization of plasma Lyso-Gb3
Lenders et al. [21]	Decrease in the left ventricular mass index

## Data Availability

Not applicable.

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
