# Peer review of "Fabry Disease: Switch from Enzyme Replacement Therapy to Oral Chaperone Migalastat: What Do We Know Today?"

_healthcare, 2023, doi:10.3390/healthcare11040449_

Round 1

Reviewer 1 Report

This manuscript reviews the literature about the switch from enzyme replacement therapy to oral chaperone migalastat in Fabry patients with amenable mutations.

The text reports correctly the paper published on the subject. However, the manuscript is inaccurate because some points are not reported or discussed enough.

1 Introduction, To explain the severity of the disease between males and females, the authors recall the Lyon hypothesis. The problem is more complex, as reported some years ago (Echevarria Clin Genetics 2016, Ref. #4). Please add this point

2 Switching from ERT to Migalastat. The description of the results of the different studies is too concise. The data on renal or cardiac parameters should be described. I suggest composing a table summarizing eGFR, proteinuria, and left ventricular mass as reported in the papers.  

3 The data by Lender (Ref.#10) on the worsening of eGFR caused many comments, and the discussion continued for a long time. The final, conclusive demonstration of the effects of migalastat on kidney function was provided by Bichet (Mol Gen Metab Reports 2021). This paper is missing. Therefore, this point is worthy of a completely new description.

4 Last point. Some limitations of Migalastat experience should be reported: the clinical experience is limited compared to ERT; the chaperone cannot be prescribed with renal failure ( eGFR<30 m/min(1,73m2). Oral therapy is more accessible to follow than intravenous, but the patient's compliance is not immediately evident. For instance, in front of a patient with no complete response to the drug, you cannot be sure of adherence to the therapy. These issues should be inserted and commented on.

Author Response

Regards.

Reviewer 2 Report

I suggest cleary reviewing  the articles that you cite   as it is unclear wich one they are describe. 

It  is necesary  carefully analize each study that is cited

Author Response

Regards.

Reviewer 3 Report

The authors have narrated the significance of migalastat in treating Fabry disease well.

But, as the review just describes the study carried out by other investigators, it does not depict any significance.

Line no: 45 instead of 'both therapies are administered'  are administered is fine. 

It will be really worthful if the author's opinion or criticism or scientific explanations are focused.

Major revision by explaining the pathophysiology of the disease and the mode of action of drugs in detail will add value to the current manuscript.

Author Response

Regards.

Round 2

Reviewer 1 Report

None

Reviewer 3 Report

Pathophysiology was not mentioned in detail but I assume it is due to the number of words constraint.